# $i$-MIX: A DOMAIN-AGNOSTIC STRATEGY FOR CONTRASTIVE REPRESENTATION LEARNING

**Kibok Lee**[1,2]  **Yian Zhu**[1]  **Kihyuk Sohn**[3]  **Chun-Liang Li**[3]  **Jinwoo Shin**[4]  **Honglak Lee**[1,5]

[1]University of Michigan  [2]Amazon Web Services  [3]Google Cloud AI  [4]KAIST  [5]LG AI Research
[1]{kibok,yianz,honglak}@umich.edu  [3]{kihyuks,chunliang}@google.com
[2]kibok@amazon.com  [4]jinwoos@kaist.ac.kr  [5]honglak@lgresearch.ai

## ABSTRACT

Contrastive representation learning has shown to be effective to learn representations from unlabeled data. However, much progress has been made in vision domains relying on data augmentations carefully designed using domain knowledge. In this work, we propose $i$-Mix, a simple yet effective domain-agnostic regularization strategy for improving contrastive representation learning. We cast contrastive learning as training a non-parametric classifier by assigning a unique virtual class to each data in a batch. Then, data instances are mixed in both the input and virtual label spaces, providing more augmented data during training. In experiments, we demonstrate that $i$-Mix consistently improves the quality of learned representations across domains, including image, speech, and tabular data. Furthermore, we confirm its regularization effect via extensive ablation studies across model and dataset sizes. The code is available at https://github.com/kibok90/imix.

## 1 INTRODUCTION

Representation learning (Bengio et al., 2013) is a fundamental task in machine learning since the success of machine learning relies on the quality of representation. Self-supervised representation learning (SSL) has been successfully applied in several domains, including image recognition (He et al., 2020; Chen et al., 2020a), natural language processing (Mikolov et al., 2013; Devlin et al., 2018), robotics (Sermanet et al., 2018; Lee et al., 2019), speech recognition (Ravanelli et al., 2020), and video understanding (Korbar et al., 2018; Owens & Efros, 2018). Since no label is available in the unsupervised setting, pretext tasks are proposed to provide self-supervision: for example, context prediction (Doersch et al., 2015), inpainting (Pathak et al., 2016), and contrastive learning (Wu et al., 2018b; Hjelm et al., 2019; He et al., 2020; Chen et al., 2020a). SSL has also been used as an auxiliary task to improve the performance on the main task, such as generative model learning (Chen et al., 2019), semi-supervised learning (Zhai et al., 2019), and improving robustness and uncertainty (Hendrycks et al., 2019).

Recently, contrastive representation learning has gained increasing attention by showing state-of-the-art performance in SSL for large-scale image recognition (He et al., 2020; Chen et al., 2020a), which outperforms its supervised pre-training counterpart (He et al., 2016) on downstream tasks. However, while the concept of contrastive learning is applicable to any domains, the quality of learned representations rely on the domain-specific inductive bias: as anchors and positive samples are obtained from the same data instance, data augmentation introduces semantically meaningful variance for better generalization. To achieve a strong, yet semantically meaningful data augmentation, domain knowledge is required, e.g., color jittering in 2D images or structural information in video understanding. Hence, contrastive representation learning in different domains requires an effort to develop effective data augmentations. Furthermore, while recent works have focused on large-scale settings where millions of unlabeled data is available, it would not be practical in real-world applications. For example, in lithography, acquiring data is very expensive in terms of both time and cost due to the complexity of manufacturing process (Lin et al., 2018; Sim et al., 2019).

Meanwhile, MixUp (Zhang et al., 2018) has shown to be a successful data augmentation for supervised learning in various domains and tasks, including image classification (Zhang et al., 2018), generative model learning (Lucas et al., 2018), and natural language processing (Guo et al., 2019; Guo, 2020).

In this paper, we explore the following natural, yet important question: is the idea of MixUp useful for unsupervised, self-supervised, or contrastive representation learning across different domains?

To this end, we propose *instance Mix (i-Mix)*, a domain-agnostic regularization strategy for contrastive representation learning. The key idea of $i$-Mix is to introduce virtual labels in a batch and mix data instances and their corresponding virtual labels in the input and label spaces, respectively. We first introduce the general formulation of $i$-Mix, and then we show the applicability of $i$-Mix to state-of-the-art contrastive representation learning methods, SimCLR (Chen et al., 2020a) and MoCo (He et al., 2020), and a self-supervised learning method without negative pairs, BYOL (Grill et al., 2020).

Through the experiments, we demonstrate the efficacy of $i$-Mix in a variety of settings. First, we show the effectiveness of $i$-Mix by evaluating the discriminative performance of learned representations in multiple domains. Specifically, we adapt $i$-Mix to the contrastive representation learning methods, advancing state-of-the-art performance across different domains, including image (Krizhevsky & Hinton, 2009; Deng et al., 2009), speech (Warden, 2018), and tabular (Asuncion & Newman, 2007) datasets. Then, we study $i$-Mix in various conditions, including when 1) the model and training dataset is small or large, 2) domain knowledge is limited, and 3) transfer learning.

**Contribution.** In summary, our contribution is three-fold:

- We propose $i$-Mix, a method for regularizing contrastive representation learning, motivated by MixUp (Zhang et al., 2018). We show how to apply $i$-Mix to state-of-the-art contrastive representation learning methods (Chen et al., 2020a; He et al., 2020; Grill et al., 2020).

- We show that $i$-Mix consistently improves contrastive representation learning in both vision and non-vision domains. In particular, the discriminative performance of representations learned with $i$-Mix is on par with fully supervised learning on CIFAR-10/100 (Krizhevsky & Hinton, 2009) and Speech Commands (Warden, 2018).

- We verify the regularization effect of $i$-Mix in a variety of settings. We empirically observed that $i$-Mix significantly improves contrastive representation learning when 1) the training dataset size is small, or 2) the domain knowledge for data augmentations is not enough.

## 2 RELATED WORK

**Self-supervised representation learning (SSL)** aims at learning representations from unlabeled data by solving a pretext task that is derived from self-supervision. Early works on SSL proposed pretext tasks based on data reconstruction by autoencoding (Bengio et al., 2007), such as context prediction (Doersch et al., 2015) and inpainting (Pathak et al., 2016). Decoder-free SSL has made a huge progress in recent years. Exemplar CNN (Dosovitskiy et al., 2014) learns by classifying individual instances with data augmentations. SSL of visual representation, including colorization (Zhang et al., 2016), solving jigsaw puzzles (Noroozi & Favaro, 2016), counting the number of objects (Noroozi et al., 2017), rotation prediction (Gidaris et al., 2018), next pixel prediction (Oord et al., 2018; Hénaff et al., 2019), and combinations of them (Doersch & Zisserman, 2017; Kim et al., 2018; Noroozi et al., 2018) often leverages image-specific properties to design pretext tasks. Meanwhile, although deep clustering (Caron et al., 2018; 2019; Asano et al., 2020) is often distinguished from SSL, it also leverages unsupervised clustering assignments as self-supervision for representation learning.

**Contrastive representation learning** has gained lots of attention for SSL (He et al., 2020; Chen et al., 2020a). As opposed to early works on exemplar CNN (Dosovitskiy et al., 2014; 2015), contrastive learning maximizes similarities of positive pairs while minimizes similarities of negative pairs instead of training an instance classifier. As the choice of negative pairs is crucial for the quality of learned representations, recent works have carefully designed them. Memory-based approaches (Wu et al., 2018b; Hjelm et al., 2019; Bachman et al., 2019; Misra & van der Maaten, 2020; Tian et al., 2020a) maintain a memory bank of embedding vectors of instances to keep negative samples, where the memory is updated with embedding vectors extracted from previous batches. In addition, MoCo (He et al., 2020) showed that differentiating the model for anchors and positive/negative samples is effective, where the model for positive/negative samples is updated by the exponential moving average of the model for anchors. On the other hand, recent works (Ye et al., 2019; Misra & van der Maaten, 2020; Chen et al., 2020a; Tian et al., 2020a) showed that learning invariance to different views is important in contrastive representation learning. The views can be generated through data augmentations carefully designed using domain knowledge (Chen et al., 2020a), splitting

input channels (Tian et al., 2020a), or borrowing the idea of other pretext tasks, such as creating jigsaw puzzles or rotating inputs (Misra & van der Maaten, 2020). In particular, SimCLR (Chen et al., 2020a) showed that a simple memory-free approach with a large batch size and strong data augmentations has a comparable performance to memory-based approaches. InfoMin (Tian et al., 2020b) further studied a way to generate good views for contrastive representation learning and achieved state-of-the-art performance by combining prior works. Different from other contrastive representation learning methods, BYOL (Grill et al., 2020) does not require negative pairs, where the proposed pretext task aims at predicting latent representations of one view from another. While prior works have focused on SSL on large-scale visual recognition tasks, our work focuses on contrastive representation learning in both small- and large-scale settings in different domains.

**Data augmentation** is a technique to increase the diversity of data, especially when training data are not enough for generalization. Since the augmented data must be understood as the original data, data augmentations are carefully designed using the domain knowledge about images (DeVries & Taylor, 2017b; Cubuk et al., 2019a;b; Zhong et al., 2020), speech (Amodei et al., 2016; Park et al., 2019), or natural languages (Zhang et al., 2015; Wei & Zou, 2019).

Some works have studied data augmentation with less domain knowledge: DeVries & Taylor (2017a) proposed a domain-agnostic augmentation strategy by first encoding the dataset and then applying augmentations in the feature space. MixUp (Zhang et al., 2018) is an effective data augmentation strategy in supervised learning, which performs vicinal risk minimization instead of empirical risk minimization, by linearly interpolating input data and their labels on the data and label spaces, respectively. On the other hand, MixUp has also shown its effectiveness in other tasks and non-vision domains, including generative adversarial networks (Lucas et al., 2018), improved robustness and uncertainty (Hendrycks et al., 2020), and sentence classification in natural language processing (Guo, 2020; Guo et al., 2019). Other variations have also been investigated by interpolating in the feature space (Verma et al., 2019) or leveraging domain knowledge (Yun et al., 2019). MixUp would not be directly applicable to some domains, such as point clouds, but its adaptation can be effective (Harris et al., 2020). $i$-Mix is a kind of data augmentation for better generalization in contrastive representation learning, resulting in better performances on downstream tasks.

**Concurrent works** have leveraged the idea of MixUp for contrastive representation learning. As discussed in Section 3.3, only input data can be mixed for improving contrastive representation learning (Shen et al., 2020; Verma et al., 2020; Zhou et al., 2020), which can be considered as injecting data-driven noises. Kalantidis et al. (2020) mixed hard negative samples on the embedding space. Kim et al. (2020) reported similar observations to ours but focused on small image datasets.

## 3 APPROACH

In this section, we review MixUp (Zhang et al., 2018) in supervised learning and present $i$-Mix in contrastive learning (He et al., 2020; Chen et al., 2020a; Grill et al., 2020). Throughout this section, let $\mathcal{X}$ be a data space, $\mathbb{R}^D$ be a $D$-dimensional embedding space, and a model $f : \mathcal{X} \to \mathbb{R}^D$ be a mapping between them. For conciseness, $f_i = f(x_i)$ and $\tilde{f}_i = f(\tilde{x}_i)$ for $x_i, \tilde{x}_i \in \mathcal{X}$, and model parameters are omitted in loss functions.

### 3.1 MIXUP IN SUPERVISED LEARNING

Suppose an one-hot label $y_i \in \{0, 1\}^C$ is assigned to a data $x_i$, where $C$ is the number of classes. Let a linear classifier predicting the labels consists of weight vectors $\{w_1, \dots, w_C\}$, where $w_c \in \mathbb{R}^D$.[1] Then, the cross-entropy loss for supervised learning is defined as:

$$\ell_{\text{Sup}}(x_i, y_i) = -\sum_{c=1}^{C} y_{i,c} \log \frac{\exp(w_c^\top f_i)}{\sum_{k=1}^{C} \exp(w_k^\top f_i)}. \tag{1}$$

While the cross-entropy loss is widely used for supervised training of deep neural networks, there are several challenges of training with the cross-entropy loss, such as preventing overfitting or networks being overconfident. Several regularization techniques have been proposed to alleviate these issues, including label smoothing (Szegedy et al., 2016), adversarial training (Miyato et al., 2018), and confidence calibration (Lee et al., 2018).

---

[1]We omit bias terms for presentation clarity.

MixUp (Zhang et al., 2018) is an effective regularization with negligible computational overhead. It conducts a linear interpolation of two data instances in both input and label spaces and trains a model by minimizing the cross-entropy loss defined on the interpolated data and labels. Specifically, for two labeled data $(x_i, y_i)$, $(x_j, y_j)$, the MixUp loss is defined as follows:

$$\ell_{\text{Sup}}^{\text{MixUp}}\big((x_i, y_i), (x_j, y_j); \lambda\big) = \ell_{\text{Sup}}(\lambda x_i + (1 - \lambda)x_j, \lambda y_i + (1 - \lambda)y_j), \qquad (2)$$

where $\lambda \sim \text{Beta}(\alpha, \alpha)$ is a mixing coefficient sampled from the beta distribution. MixUp is a vicinal risk minimization method (Chapelle et al., 2001) that augments data and their labels in a data-driven manner. Not only improving the generalization on the supervised task, it also improves adversarial robustness (Pang et al., 2019) and confidence calibration (Thulasidasan et al., 2019).

### 3.2 $i$-MIX IN CONTRASTIVE LEARNING

We introduce *instance mix (i-Mix)*, a data-driven augmentation strategy for contrastive representation learning to improve the generalization of learned representations. Intuitively, instead of mixing class labels, $i$-Mix interpolates their *virtual* labels, which indicates their identity in a batch.

Let $\mathcal{B} = \{(x_i, \tilde{x}_i)\}_{i=1}^N$ be a batch of data pairs, where $N$ is the batch size, $x_i, \tilde{x}_i \in \mathcal{X}$ are two views of the same data, which are usually generated by different augmentations. For each anchor $x_i$, we call $\tilde{x}_i$ and $\tilde{x}_{j \neq i}$ positive and negative samples, respectively.[2] Then, the model $f$ learns to maximize similarities of positive pairs (instances from the same data) while minimize similarities of negative pairs (instances from different data) in the embedding space. The output of $f$ is L2-normalized, which has shown to be effective (Wu et al., 2018a; He et al., 2020; Chen et al., 2020a). Let $v_i \in \{0, 1\}^N$ be the virtual label of $x_i$ and $\tilde{x}_i$ in a batch $\mathcal{B}$, where $v_{i,i} = 1$ and $v_{i,j \neq i} = 0$. For a general sample-wise contrastive loss with virtual labels $\ell(x_i, v_i)$, the $i$-Mix loss is defined as follows:

$$\ell^{i\text{-Mix}}\big((x_i, v_i), (x_j, v_j); \mathcal{B}, \lambda\big) = \ell(\text{Mix}(x_i, x_j; \lambda), \lambda v_i + (1 - \lambda)v_j; \mathcal{B}), \qquad (3)$$

where $\lambda \sim \text{Beta}(\alpha, \alpha)$ is a mixing coefficient and Mix is a mixing operator, which can be adapted depending on target domains: for example, $\text{MixUp}(x_i, x_j; \lambda) = \lambda x_i + (1 - \lambda)x_j$ (Zhang et al., 2018) when data values are continuous, and $\text{CutMix}(x_i, x_j; \lambda) = M_\lambda \odot x_i + (1 - M_\lambda) \odot x_j$ (Yun et al., 2019) when data values have a spatial correlation with neighbors, where $M_\lambda$ is a binary mask filtering out a region whose relative area is $(1 - \lambda)$, and $\odot$ is an element-wise multiplication. Note that some mixing operators might not work well for some domains: for example, CutMix would not be valid when data values and their spatial neighbors have no correlation. However, the MixUp operator generally works well across domains including image, speech, and tabular; we use it for $i$-Mix formulations and experiments, unless otherwise specified. In the following, we show how to apply $i$-Mix to contrastive representation learning methods.

**SimCLR** (Chen et al., 2020a) is a simple contrastive representation learning method without a memory bank, where each anchor has one positive sample and $(2N-2)$ negative samples. Let $x_{N+i} = \tilde{x}_i$ for conciseness. Then, the $(2N-1)$-way discrimination loss is written as follows:

$$\ell_{\text{SimCLR}}(x_i; \mathcal{B}) = -\log \frac{\exp\big(s(f_i, f_{(N+i) \bmod 2N})/\tau\big)}{\sum_{k=1, k \neq i}^{2N} \exp\big(s(f_i, f_k)/\tau\big)}, \qquad (4)$$

where $\tau$ is a temperature scaling parameter and $s(f, \tilde{f}) = (f^\top \tilde{f})/\|f\|\|\tilde{f}\|$ is the inner product of two L2-normalized vectors. In this formulation, $i$-Mix is not directly applicable because virtual labels are defined differently for each anchor.[3] To resolve this issue, we simplify the formulation of SimCLR by excluding anchors from negative samples. Then, with virtual labels, the $N$-way discrimination loss is written as follows:

$$\ell_{\text{N-pair}}(x_i, v_i; \mathcal{B}) = -\sum_{n=1}^N v_{i,n} \log \frac{\exp\big(s(f_i, \tilde{f}_n)/\tau\big)}{\sum_{k=1}^N \exp\big(s(f_i, \tilde{f}_k)/\tau\big)}, \qquad (5)$$

where we call it the **N-pair** contrastive loss, as the formulation is similar to the N-pair loss in the context of metric learning (Sohn, 2016).[4] For two data instances $(x_i, v_i)$, $(x_j, v_j)$ and a batch of data

---

[2]Some literature (He et al., 2020; Chen et al., 2020b) refers to them as query and positive/negative keys.

[3]We present the application of $i$-Mix to the original SimCLR formulation in Appendix A.

[4]InfoNCE (Oord et al., 2018) is a similar loss inspired by the idea of noise-contrastive estimation (Gutmann & Hyvärinen, 2010).

---

**Algorithm 1** Loss computation for $i$-Mix on N-pair contrastive learning in PyTorch-like style.

```
a, b = aug(x), aug(x) # two different views of input x
lam = Beta(alpha, alpha).sample() # mixing coefficient
randidx = randperm(len(x))
a = lam * a + (1-lam) * a[randidx]
logits = matmul(normalize(model(a)), normalize(model(b)).T) / t
loss = lam * CrossEntropyLoss(logits, arange(len(x))) + \
       (1-lam) * CrossEntropyLoss(logits, randidx)
```

---

pairs $\mathcal{B} = \{(x_i, \tilde{x}_i)\}_{i=1}^N$, the $i$-Mix loss is defined as follows:

$$\ell_{\text{N-pair}}^{i\text{-Mix}}\big((x_i, v_i), (x_j, v_j); \mathcal{B}, \lambda\big) = \ell_{\text{N-pair}}(\lambda x_i + (1-\lambda)x_j, \lambda v_i + (1-\lambda)v_j; \mathcal{B}). \tag{6}$$

Algorithm 1 provides the pseudocode of $i$-Mix on N-pair contrastive learning for one iteration.[5]

**Pair relations in contrastive loss.** To use contrastive loss for representation learning, one needs to properly define a pair relation $\{(x_i, \tilde{x}_i)\}_{i=1}^N$. For contrastive representation learning, where semantic class labels are not provided, the pair relation would be defined in that 1) a positive pair, $x_i$ and $\tilde{x}_i$, are different views of the same data and 2) a negative pair, $x_i$ and $\tilde{x}_{j \neq i}$, are different data instances. For supervised representation learning, $x_i$ and $\tilde{x}_i$ are two data instances from the same class, while $x_i$ and $\tilde{x}_{j \neq i}$ are from different classes. Note that two augmented versions of the same data also belong to the same class, so they can also be considered as a positive pair. $i$-Mix is not limited to self-supervised contrastive representation learning, but it can also be used as a regularization method for supervised contrastive representation learning (Khosla et al., 2020) or deep metric learning (Sohn, 2016; Movshovitz-Attias et al., 2017).

**MoCo (He et al., 2020).** In contrastive representation learning, the number of negative samples affects the quality of learned representations (Arora et al., 2019). Because SimCLR mines negative samples in the current batch, having a large batch size is crucial, which often requires a lot of computational resources (Chen et al., 2020a). For efficient training, recent works have maintained a memory bank $\mathcal{M} = \{\mu_k\}_{k=1}^K$, which is a queue of previously extracted embedding vectors, where $K$ is the size of the memory bank (Wu et al., 2018b; He et al., 2020; Tian et al., 2020a;b). In addition, MoCo introduces an exponential moving average (EMA) model to extract positive and negative embedding vectors, whose parameters are updated as $\theta_{f^{\text{EMA}}} \leftarrow m\theta_{f^{\text{EMA}}} + (1-m)\theta_f$, where $m \in [0, 1)$ is a momentum coefficient and $\theta$ is model parameters. The loss is written as follows:

$$\ell_{\text{MoCo}}(x_i; \mathcal{B}, \mathcal{M}) = -\log \frac{\exp\big(s(f_i, \tilde{f}_i^{\text{EMA}})/\tau\big)}{\exp\big(s(f_i, \tilde{f}_i^{\text{EMA}})/\tau\big) + \sum_{k=1}^K \exp\big(s(f_i, \mu_k)/\tau\big)}. \tag{7}$$

The memory bank $\mathcal{M}$ is then updated with $\{\tilde{f}_i^{\text{EMA}}\}$ in the first-in first-out order. In this $(K+1)$-way discrimination loss, data pairs are independent to each other, such that $i$-Mix is not directly applicable because virtual labels are defined differently for each anchor. To overcome this issue, we include the positive samples of other anchors as negative samples, similar to the N-pair contrastive loss in Eq. (5). Let $\tilde{v}_i \in \{0, 1\}^{N+K}$ be a virtual label indicating the positive sample of each anchor, where $\tilde{v}_{i,i} = 1$ and $\tilde{v}_{i,j \neq i} = 0$. Then, the $(N+K)$-way discrimination loss is written as follows:

$$\ell_{\text{MoCo}}(x_i, \tilde{v}_i; \mathcal{B}, \mathcal{M}) = -\sum_{n=1}^N \tilde{v}_{i,n} \log \frac{\exp\big(s(f_i, \tilde{f}_n^{\text{EMA}})/\tau\big)}{\sum_{k=1}^N \exp\big(s(f_i, \tilde{f}_k^{\text{EMA}})/\tau\big) + \sum_{k=1}^K \exp\big(s(f_i, \mu_k)/\tau\big)}. \tag{8}$$

As virtual labels are bounded in the same set in this formulation, $i$-Mix is directly applicable: for two data instances $(x_i, \tilde{v}_i), (x_j, \tilde{v}_j)$, a batch of data pairs $\mathcal{B} = \{(x_i, \tilde{x}_i)\}_{i=1}^N$, and the memory bank $\mathcal{M}$, the $i$-Mix loss is defined as follows:

$$\ell_{\text{MoCo}}^{i\text{-Mix}}\big((x_i, \tilde{v}_i), (x_j, \tilde{v}_j); \mathcal{B}, \mathcal{M}, \lambda\big) = \ell_{\text{MoCo}}(\lambda x_i + (1-\lambda)x_j, \lambda \tilde{v}_i + (1-\lambda)\tilde{v}_j; \mathcal{B}, \mathcal{M}). \tag{9}$$

---

[5] For losses linear with respect to labels (e.g., the cross-entropy loss), they are equivalent to $\lambda \ell(\lambda x_i + (1-\lambda)x_j, v_i) + (1-\lambda)\ell(\lambda x_i + (1-\lambda)x_j, v_j)$, i.e., optimization to the mixed label is equivalent to joint optimization to original labels. The proof for losses in contrastive learning methods is provided in Appendix B.

**BYOL (Grill et al., 2020).** Different from other contrastive representation learning methods, BYOL is a self-supervised representation learning method without contrasting negative pairs. For two views of the same data $x_i, \tilde{x}_i \in \mathcal{X}$, the model $f$ learns to predict a view embedded with the EMA model $\tilde{f}_i^{\text{EMA}}$ from its embedding $f_i$. Specifically, an additional prediction layer $g$ is introduced, such that the difference between $g(f_i)$ and $\tilde{f}_i^{\text{EMA}}$ is learned to be minimized. The BYOL loss is written as follows:

$$\ell_{\text{BYOL}}(x_i, \tilde{x}_i) = \left\| g(f_i)/\|g(f_i)\| - \tilde{f}_i/\|\tilde{f}_i\| \right\|^2 = 2 - 2 \cdot s(g(f_i), \tilde{f}_i). \tag{10}$$

This formulation can be represented in the form of the general contrastive loss in Eq. (3), as the second view $\tilde{x}_i$ can be accessed from the batch $\mathcal{B}$ with its virtual label $v_i$. To derive $i$-Mix in BYOL, let $\tilde{F} = [\tilde{f}_1/\|\tilde{f}_1\|, \dots, \tilde{f}_N/\|\tilde{f}_N\|] \in \mathbb{R}^{D \times N}$ be the collection of L2-normalized embedding vectors of the second views, such that $\tilde{f}_i/\|\tilde{f}_i\| = \tilde{F}v_i$. Then, the BYOL loss is written as follows:

$$\ell_{\text{BYOL}}(x_i, v_i; \mathcal{B}) = \left\| g(f_i)/\|g(f_i)\| - \tilde{F}v_i \right\|^2 = 2 - 2 \cdot s(g(f_i), \tilde{F}v_i). \tag{11}$$

For two data instances $(x_i, v_i), (x_j, v_j)$ and a batch of data pairs $\mathcal{B} = \{(x_i, \tilde{x}_i)\}_{i=1}^N$, the $i$-Mix loss is defined as follows:

$$\ell_{\text{BYOL}}^{i\text{-Mix}}\big((x_i, v_i), (x_j, v_j); \mathcal{B}, \lambda\big) = \ell_{\text{BYOL}}(\lambda x_i + (1-\lambda)x_j, \lambda v_i + (1-\lambda)v_j; \mathcal{B}). \tag{12}$$

### 3.3 INPUTMIX

The contribution of data augmentations to the quality of learned representations is crucial in contrastive representation learning. For the case when the domain knowledge about efficient data augmentations is limited, we propose to apply InputMix together with $i$-Mix, which mixes input data but not their labels. This method can be viewed as introducing structured noises driven by auxiliary data to the principal data with the largest mixing coefficient $\lambda$, and the label of the principal data is assigned to the mixed data (Shen et al., 2020; Verma et al., 2020; Zhou et al., 2020). We applied InputMix and $i$-Mix together on image datasets in Table 3.

## 4 EXPERIMENTS

In this section, we demonstrate the effectiveness of $i$-Mix. In all experiments, we conduct contrastive representation learning on a pretext dataset and evaluate the quality of representations via supervised classification on a downstream dataset. We report the accuracy averaged over up to five runs. In the first stage, a convolutional neural network (CNN) or multilayer perceptron (MLP) followed by the two-layer MLP projection head is trained on an unlabeled dataset. Then, we replace the projection head with a linear classifier and train only the linear classifier on a labeled dataset for downstream task. Except for transfer learning, datasets for the pretext and downstream tasks are the same. For $i$-Mix, we sample a mixing coefficient $\lambda \sim \text{Beta}(\alpha, \alpha)$ for each data, where $\alpha = 1$ unless otherwise stated.[6] Additional details for the experimental settings and more experiments can be found in Appendix C.

### 4.1 EXPERIMENTAL SETUP

**Baselines and datasets.** We consider 1) N-pair contrastive learning as a memory-free contrastive learning method,[7] 2) MoCo v2 (He et al., 2020; Chen et al., 2020b)[8] as a memory-based contrastive learning method, and 3) BYOL (Grill et al., 2020), which is a self-supervised learning method without negative pairs. We apply $i$-Mix to these methods and compare their performances. To show the effectiveness of $i$-Mix across domains, we evaluate the methods on datasets from multiple domains, including image, speech, and tabular datasets.

CIFAR-10/100 (Krizhevsky & Hinton, 2009) consist of 50k training and 10k test images, and ImageNet (Deng et al., 2009) has 1.3M training and 50k validation images, where we use them for evaluation. For ImageNet, we also use a subset of randomly chosen 100 classes out of 1k classes to experiment at a different scale. We apply a set of data augmentations randomly in sequence including

---

[6] $\text{Beta}(\alpha, \alpha)$ is the uniform distribution when $\alpha = 1$, bell-shaped when $\alpha > 1$, and bimodal when $\alpha < 1$.

[7] We use the N-pair formulation in Eq. (5) instead of SimCLR as it is simpler and more efficient to integrate $i$-Mix. As shown in Appendix C.2, the N-pair formulation results in no worse performance than SimCLR.

[8] MoCo v2 improves the performance of MoCo by cosine learning schedule and more data augmentations.

| Domain | Dataset | N-pair | + $i$-Mix | MoCo v2 | + $i$-Mix | BYOL | + $i$-Mix |
|--------|---------|--------|-----------|---------|-----------|------|-----------|
| Image | CIFAR-10 | 93.3 ± 0.1 | **95.6** ± 0.2 | 93.5 ± 0.2 | **96.1** ± 0.1 | 94.2 ± 0.2 | **96.3** ± 0.2 |
| | CIFAR-100 | 70.8 ± 0.4 | **75.8** ± 0.3 | 71.6 ± 0.1 | **78.1** ± 0.3 | 72.7 ± 0.4 | **78.6** ± 0.2 |
| Speech | Commands | 94.9 ± 0.1 | **98.3** ± 0.1 | 96.3 ± 0.1 | **98.4** ± 0.0 | 94.8 ± 0.2 | **98.3** ± 0.0 |
| Tabular | CovType | 68.5 ± 0.3 | **72.1** ± 0.2 | 70.5 ± 0.2 | **73.1** ± 0.1 | 72.1 ± 0.2 | **74.1** ± 0.2 |

Table 1: Comparison of contrastive representation learning methods and $i$-Mix in different domains.

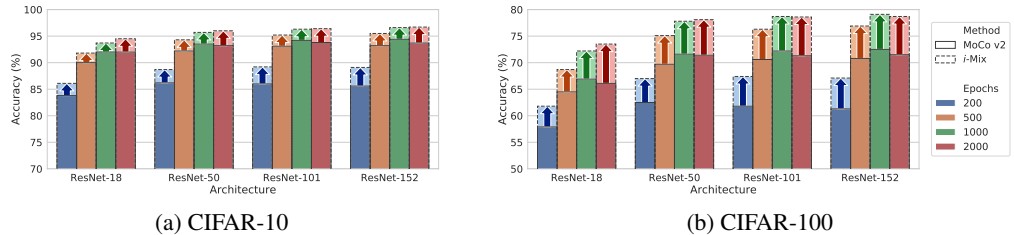

(a) CIFAR-10                    (b) CIFAR-100

Figure 1: Comparison of performance gains by applying $i$-Mix to MoCo v2 with different model sizes and number of epochs on CIFAR-10 and 100.

random resized cropping, horizontal flipping, color jittering, gray scaling, and Gaussian blurring for ImageNet, which has shown to be effective (Chen et al., 2020a;b). We use ResNet-50 (He et al., 2016) as a backbone network. Models are trained with a batch size of 256 (i.e., 512 including augmented data) for up to 4000 epochs on CIFAR-10 and 100, and with a batch size of 512 for 800 epochs on ImageNet. For ImageNet experiments, we use the CutMix (Yun et al., 2019) version of $i$-Mix.

The Speech Commands dataset (Warden, 2018) contains 51k training, 7k validation, and 7k test data in 12 classes. We apply a set of data augmentations randomly in sequence: changing amplitude, speed, and pitch in time domain, stretching, time shifting, and adding background noise in frequency domain. Augmented data are then transformed to a 32×32 mel spectogram. We use the same architecture with image experiments. Models are trained with a batch size of 256 for 500 epochs.

For tabular dataset experiments, we consider Forest Cover Type (CovType) and Higgs Boson (Higgs) from UCI repository (Asuncion & Newman, 2007). CovType contains 15k training and 566k test data in 7 classes, and Higgs contains 10.5M training and 0.5M test data for binary classification. For Higgs, we use a subset of 100k and 1M training data to experiment at a different scale. Since the domain knowledge for data augmentations on tabular data is limited, only a masking noise with the probability 0.2 is considered as a data augmentation. We use a 5-layer MLP with batch normalization (Ioffe & Szegedy, 2015) as a backbone network. Models are trained with a batch size of 512 for 500 epochs. We use $\alpha = 2$ for CovType and Higgs100k, as it is slightly better than $\alpha = 1$.

## 4.2 MAIN RESULTS

Table 1 shows the wide applicability of $i$-Mix to state-of-the-art contrastive representation learning methods in multiple domains. $i$-Mix results in consistent improvements on the classification accuracy, e.g., up to 6.5% when $i$-Mix is applied to MoCo v2 on CIFAR-100. Interestingly, we observe that linear classifiers on top of representations learned with $i$-Mix without fine-tuning the pre-trained part often yield a classification accuracy on par with simple end-to-end supervised learning from random initialization, e.g., $i$-Mix vs. end-to-end supervised learning performance is 96.3% vs. 95.5% on CIFAR-10, 78.6% vs. 78.9% on CIFAR-100, and 98.2% vs. 98.0% on Speech Commands.[9]

## 4.3 REGULARIZATION EFFECT OF $i$-MIX

A better regularization method often benefits from longer training of deeper models, which is more critical when training on a small dataset. To investigate the regularization effect of $i$-Mix, we first

---

[9]Supervised learning with improved methods, e.g., MixUp, outperforms $i$-Mix. However, linear evaluation on top of self-supervised representation learning is a proxy to measure the quality of representations learned without labels, such that it is not supposed to be compared with the performance of supervised learning.

| Domain | Dataset | MoCo v2 | + $i$-Mix |
|--------|---------|---------|-----------|
| Image | ImageNet-100 | 84.1 | **87.0** |
| | ImageNet-1k | 70.9 | **71.3** |

| Domain | Dataset | MoCo v2 | + $i$-Mix |
|--------|---------|---------|-----------|
| Tabular | Higgs100k | 72.1 | **72.9** |
| | Higgs1M | **74.9** | 74.5 |

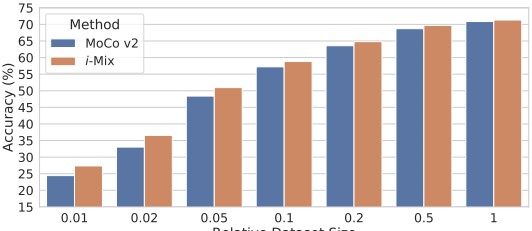

Table 2: Comparison of MoCo v2 and $i$-Mix on large-scale datasets.

Figure 2: Comparison of MoCo v2 and $i$-Mix trained on the different size of ImageNet.

| Aug | CIFAR-10 | | CIFAR-100 | | Speech Commands | | CovType | | Higgs100k | | Higgs1M | |
|-----|----------|----------|-----------|----------|-----------------|----------|---------|----------|-----------|----------|---------|----------|
| | MoCo v2 | + $i$-Mix* | MoCo v2 | + $i$-Mix* | MoCo v2 | + $i$-Mix | MoCo v2 | + $i$-Mix | MoCo v2 | + $i$-Mix | MoCo v2 | + $i$-Mix |
| - | 47.7 ± 1.3 | **83.4** ± 0.4 | 24.7 ± 0.7 | **54.0** ± 0.5 | 76.9 ± 1.7 | **92.8** ± 0.5 | 69.6 ± 0.3 | **73.1** ± 0.1 | 64.2 | **71.8** | 65.5 | **72.9** |
| ✓ | 93.5 ± 0.2 | **96.1** ± 0.1 | 71.6 ± 0.1 | **78.1** ± 0.3 | 96.3 ± 0.1 | **98.4** ± 0.0 | 70.5 ± 0.2 | **73.1** ± 0.1 | 72.1 | **72.9** | **74.9** | 74.5 |

Table 3: Comparison of MoCo v2 and $i$-Mix with and without data augmentations.

*InputMix is applied when no other data augmentations are used.

make a comparison between MoCo v2 and $i$-Mix by training with different model sizes and number of training epochs on the pretext task. We train ResNet-18, 50, 101, and 152 models with varying number of training epochs from 200 to 2000. Figure 1 shows the performance of MoCo v2 (solid box) and $i$-Mix (dashed box). The improvement by applying $i$-Mix to MoCo v2 is consistent over the different architecture size and the number of training epochs. Deeper models benefit from $i$-Mix, achieving 96.7% on CIFAR-10 and 79.1% on CIFAR-100 when the backbone network is ResNet-152. On the other hand, models trained without $i$-Mix start to show decrease in performance, possibly due to overfitting to the pretext task when trained longer. The trend clearly shows that $i$-Mix results in better representations via improved regularization.

Next, we study the effect of $i$-Mix with varying dataset sizes for the pretext tasks. Table 2 shows the effect of $i$-Mix on large-scale datasets[10] from image and tabular domains. We observe that $i$-Mix is particularly effective when the amount of training data is reduced, e.g., ImageNet-100 consists of images from 100 classes, thus has only 10% of training data compared to ImageNet-1k. However, the performance gain is reduced when the amount of training data is large. we further study representations learned with different pretext dataset sizes from 1% to 100% of the ImageNet training data in Figure 2. Here, different from ImageNet-100, we reduce the amount of data for each class, but maintain the number of classes the same. We observe that the performance gain by $i$-Mix is more significant when the size of the pretext dataset is small. Our study suggests that $i$-Mix is effective for regularizing self-supervised representation learning when training from a limited amount of data. We believe that this is aligned with findings in Zhang et al. (2018) for MixUp in supervised learning. Finally, when a large-scale unlabeled dataset is available, we expect $i$-Mix would still be useful in obtaining better representations when trained longer with deeper and larger models.

### 4.4 CONTRASTIVE LEARNING WITHOUT DOMAIN-SPECIFIC DATA AUGMENTATION

Data augmentations play a key role in contrastive representation learning, and therefore it raises a question when applying them to domains with a limited or no knowledge of such augmentations. In this section, we study the effectiveness of $i$-Mix as a domain-agnostic strategy for contrastive representation learning, which can be adapted to different domains. Table 3 shows the performance of MoCo v2 and $i$-Mix with and without data augmentations. We observe significant performance gains with $i$-Mix when other data augmentations are not applied. For example, compared to the accuracy of 93.5% on CIFAR-10 when other data augmentations are applied, contrastive learning achieves 47.7% when trained without any data augmentations. This suggests that data augmentation is an essential part for the success of contrastive representation learning (Chen et al., 2020a). However, $i$-Mix is able to learn meaningful representations without other data augmentations and achieves the accuracy of 83.4% on CIFAR-10.

---

[10]Here, "scale" corresponds to the amount of data rather than image resolution.

| Pretext | CIFAR-10 | | CIFAR-100 | |
|---|---|---|---|---|
| Downstream | MoCo v2 | + $i$-Mix | MoCo v2 | + $i$-Mix |
| CIFAR-10 | $93.5 \pm 0.2$ | $\mathbf{96.1} \pm 0.1$ | $85.9 \pm 0.3$ | $\mathbf{90.0} \pm 0.4$ |
| CIFAR-100 | $64.1 \pm 0.4$ | $\mathbf{70.8} \pm 0.4$ | $71.6 \pm 0.1$ | $\mathbf{78.1} \pm 0.3$ |

(a) CIFAR-10 and 100 as the pretext dataset

| VOC Object Detection | ImageNet | |
|---|---|---|
| | MoCo v2 | + $i$-Mix |
| AP | $57.3 \pm 0.1$ | $\mathbf{57.5} \pm 0.4$ |
| $AP_{50}$ | $82.5 \pm 0.2$ | $\mathbf{82.7} \pm 0.2$ |
| $AP_{75}$ | $63.8 \pm 0.3$ | $\mathbf{64.2} \pm 0.7$ |

(b) ImageNet as the pretext dataset

Table 4: Comparison of MoCo v2 and $i$-Mix in transfer learning.

In Table 3, InputMix is applied together with $i$-Mix to further improve the performance on image datasets. For each principal data, we mix two auxiliary data, with mixing coefficients $(0.5\lambda_1 + 0.5, 0.5\lambda_2, 0.5\lambda_3)$, where $\lambda_1, \lambda_2, \lambda_3 \sim \text{Dirichlet}(1, 1, 1)$.[11] In the above example, while $i$-Mix is better than baselines, adding InputMix further improves the performance of $i$-Mix, i.e., from 75.1% to 83.4% on CIFAR-10, and from 50.7% to 54.0% on CIFAR-100. This confirms that InputMix can further improve the performance when domain-specific data augmentations are not available, as discussed in Section 3.3.

Moreover, we verify its effectiveness on other domains beyond the image domain. For example, the performance improves from 76.9% to 92.8% on the Speech Commands dataset when we assume no other data augmentations are available. We also observe consistent improvements in accuracy for tabular datasets, even when the training dataset size is large. Although the domain knowledge for data augmentations is important to achieve state-of-the-art results, our demonstration shows the potential of $i$-Mix to be used for a wide range of application domains where domain knowledge is particularly limited.

## 4.5 TRANSFERABILITY OF $i$-MIX

In this section, we show the improved transferability of the representations learned with $i$-Mix. The results are provided in Table 4. First, we train linear classifiers with downstream datasets different from the pretext dataset used to train backbone networks and evaluate their performance, e.g., CIFAR-10 as pretext and CIFAR-100 as downstream datasets or vice versa. We observe consistent performance gains when learned representations from one dataset are evaluated on classification tasks of another dataset. Next, we transfer representations trained on ImageNet to the PASCAL VOC object detection task (Everingham et al., 2010). We follow the settings in prior works (He et al., 2020; Chen et al., 2020b): the parameters of the pre-trained ResNet-50 are transferred to a Faster R-CNN detector with the ResNet50-C4 backbone (Ren et al., 2015), and fine-tuned end-to-end on the VOC 07+12 trainval dataset and evaluated on the VOC 07 test dataset. We report the average precision (AP) averaged over IoU thresholds between 50% to 95% at a step of 5%, and $AP_{50}$ and $AP_{75}$, which are AP values when IoU threshold is 50% and 75%, respectively. Similar to Table 2, we observe small but consistent performance gains in all metrics. Those results confirm that $i$-Mix improves the quality of learned representations, such that performances on downstream tasks are improved.

## 5 CONCLUSION

We propose $i$-Mix, a domain-agnostic regularization strategy applicable to a class of self-supervised learning. The key idea of $i$-Mix is to introduce a virtual label to each data instance, and mix both inputs and the corresponding virtual labels. We show that $i$-Mix is applicable to state-of-the-art self-supervised representation learning methods including SimCLR, MoCo, and BYOL, which consistently improves the performance in a variety of settings and domains. Our experimental results indicate that $i$-Mix is particularly effective when the training dataset size is small or data augmentation is not available, each of which are prevalent in practice.

---

[11]This guarantees that the mixing coefficient for the principal data is larger than 0.5 to prevent from training with noisy labels. Note that Beckham et al. (2019) also sampled mixing coefficients from the Dirichlet distribution for mixing more than two data.

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

# A   MORE APPLICATIONS OF $i$-MIX

In this section, we introduce more variations of $i$-Mix. For conciseness, we use $v_i$ to denote virtual labels for different methods. We make the definition of $v_i$ for each application clear.

## A.1   $i$-MIX FOR SIMCLR

For each anchor, SimCLR takes other anchors as negative samples such that the virtual labels must be extended. Let $x_{N+i} = \tilde{x}_i$ for conciseness, and $v_i \in \{0, 1\}^{2N}$ be the virtual label indicating the positive sample of each anchor, where $v_{i,N+i} = 1$ and $v_{i,j \neq N+i} = 0$. Note that $v_{i,i} = 0$ because the anchor itself is not counted as a positive sample. Then, Eq. (4) can be represented in the form of the cross-entropy loss:

$$\ell_{\text{SimCLR}}(x_i, v_i; \mathcal{B}) = -\sum_{n=1}^{2N} v_{i,n} \log \frac{\exp\big(s(f_i, f_n)/\tau\big)}{\sum_{k=1, k\neq i}^{2N} \exp\big(s(f_i, f_k)/\tau\big)}. \tag{A.1}$$

The application of $i$-Mix to SimCLR is straightforward: for two data instances $(x_i, v_i)$, $(x_j, v_j)$ and a batch of data $\mathcal{B} = \{x_i\}_{i=1}^{2N}$, the $i$-Mix loss is defined as follows:[12]

$$\ell_{\text{SimCLR}}^{i\text{-Mix}}\big((x_i, v_i), (x_j, v_j); \mathcal{B}, \lambda\big) = \ell_{\text{SimCLR}}(\lambda x_i + (1-\lambda)x_j, \lambda v_i + (1-\lambda)v_j; \mathcal{B}). \tag{A.2}$$

Note that only the input data of Eq. (A.2) is mixed, such that $f_i$ in Eq. (A.1) is an embedding vector of the mixed data while the other $f_n$'s are the ones of clean data. Because both clean and mixed data need to be fed to the network $f$, $i$-Mix for SimCLR requires twice more memory and training time compared to SimCLR when the same batch size is used.

## A.2   $i$-MIX FOR SUPERVISED CONTRASTIVE LEARNING

Supervised contrastive learning has recently shown to be effective for supervised representation learning and it often outperforms the standard end-to-end supervised classifier learning (Khosla et al., 2020). Suppose an one-hot label $y_i \in \{0, 1\}^C$ is assigned to a data $x_i$, where $C$ is the number of classes. Let $x_{N+i} = \tilde{x}_i$ and $y_{N+i} = y_i$ for conciseness. For a batch of data pairs and their labels $\mathcal{B} = \{(x_i, y_i)\}_{i=1}^{2N}$, let $v_i \in \{0, 1\}^{2N}$ be the virtual label indicating the positive samples of each anchor, where $v_{i,j} = 1$ if $y_i = y_{j \neq i}$, and otherwise $v_{i,j} = 0$. Intuitively, $\sum_{j=1}^{2N} v_{i,j} = 2N_{y_i} - 1$ where $N_{y_i}$ is the number of data with the label $y_i$. Then, the supervised learning version of the SimCLR (SupCLR) loss function is written as follows:

$$\ell_{\text{SupCLR}}(x_i, v_i; \mathcal{B}) = -\frac{1}{2N_{y_i} - 1} \sum_{n=1}^{2N} v_{i,n} \log \frac{\exp\big(s(f_i, f_n)/\tau\big)}{\sum_{k=1, k\neq i}^{2N} \exp\big(s(f_i, f_k)/\tau\big)}. \tag{A.3}$$

The application of $i$-Mix to SupCLR is straightforward: for two data instances $(x_i, v_i)$, $(x_j, v_j)$ and a batch of data $\mathcal{B} = \{x_i\}_{i=1}^{2N}$, the $i$-Mix loss is defined as follows:

$$\ell_{\text{SupCLR}}^{i\text{-Mix}}\big((x_i, v_i), (x_j, v_j); \mathcal{B}, \lambda\big) = \ell_{\text{SupCLR}}(\lambda x_i + (1-\lambda)x_j, \lambda v_i + (1-\lambda)v_j; \mathcal{B}). \tag{A.4}$$

## A.3   $i$-MIX FOR N-PAIR SUPERVISED CONTRASTIVE LEARNING

Note that $i$-Mix in Eq. (A.4) is not as efficient as SupCLR in Eq. (A.3) due to the same reason in the case of SimCLR. To overcome this, we reformulate SupCLR in the form of the N-pair loss (Sohn, 2016). Suppose an one-hot label $y_i \in \{0, 1\}^C$ is assigned to a data $x_i$, where $C$ is the number of classes. For a batch of data pairs and their labels $\mathcal{B} = \{(x_i, \tilde{x}_i, y_i)\}_{i=1}^{N}$, let $v_i \in \{0, 1\}^N$ be the virtual label indicating the positive samples of each anchor, where $v_{i,j} = 1$ if $y_i = y_{j \neq i}$, and otherwise $v_{i,j} = 0$. Then, the supervised version of the N-pair (Sup-N-pair) contrastive loss function is written as follows:

$$\ell_{\text{Sup-N-pair}}(x_i, v_i; \mathcal{B}) = -\frac{1}{N_{y_i}} \sum_{n=1}^{N} v_{i,n} \log \frac{\exp\big(s(f_i, \tilde{f}_n)/\tau\big)}{\sum_{k=1}^{N} \exp\big(s(f_i, \tilde{f}_k)/\tau\big)}. \tag{A.5}$$

Then, the $i$-Mix loss for Sup-N-pair is defined as follows:

$$\ell_{\text{Sup-N-pair}}^{i\text{-Mix}}\big((x_i, v_i), (x_j, v_j); \mathcal{B}, \lambda\big) = \ell_{\text{Sup-N-pair}}(\lambda x_i + (1-\lambda)x_j, \lambda v_i + (1-\lambda)v_j; \mathcal{B}). \tag{A.6}$$

---

[12]The $j$-th data can be excluded from the negative samples, but it does not result in a significant difference.

# B    PROOF OF THE LINEARITY OF LOSSES WITH RESPECT TO VIRTUAL LABELS

**Cross-entropy loss.** The loss used in contrastive representation learning works, which is often referred to as InfoNCE (Oord et al., 2018), can be represented in the form of the cross-entropy loss as we showed for N-pair contrastive learning, SimCLR (Chen et al., 2020a), and MoCo (He et al., 2020). Here we provide an example in the case of N-pair contrastive learning. Let $f_{ij}^\lambda = f(\lambda x_i + (1 - \lambda)x_j)$ for conciseness.

$$
\ell_{\text{N-pair}}^{i\text{-Mix}}\big((x_i, v_i), (x_j, v_j); \mathcal{B}, \lambda\big) = \ell_{\text{N-pair}}(\lambda x_i + (1 - \lambda)x_j, \lambda v_i + (1 - \lambda)v_j; \mathcal{B})
$$

$$
= -\sum_{n=1}^{N}(\lambda v_{i,n} + (1 - \lambda)v_{j,n}) \log \frac{\exp\big(s(f_{ij}^\lambda, \tilde{f}_n)/\tau\big)}{\sum_{k=1}^{N} \exp\big(s(f_{ij}^\lambda, \tilde{f}_k)/\tau\big)}
$$

$$
= -\lambda \sum_{n=1}^{N} v_{i,n} \log \frac{\exp\big(s(f_{ij}^\lambda, \tilde{f}_n)/\tau\big)}{\sum_{k=1}^{N} \exp\big(s(f_{ij}^\lambda, \tilde{f}_k)/\tau\big)} - (1 - \lambda) \sum_{n=1}^{N} v_{j,n} \log \frac{\exp\big(s(f_{ij}^\lambda, \tilde{f}_n)/\tau\big)}{\sum_{k=1}^{N} \exp\big(s(f_{ij}^\lambda, \tilde{f}_k)/\tau\big)}
$$

$$
= \lambda \ell_{\text{N-pair}}(\lambda x_i + (1 - \lambda)x_j, v_i; \mathcal{B}) + (1 - \lambda)\ell_{\text{N-pair}}(\lambda x_i + (1 - \lambda)x_j, v_j; \mathcal{B}). \tag{B.1}
$$

**L2 loss between L2-normalized feature vectors.** The BYOL (Grill et al., 2020) loss is in this type. Let $\tilde{F} = [\tilde{f}_1/\|\tilde{f}_1\|, \ldots, \tilde{f}_N/\|\tilde{f}_N\|] \in \mathbb{R}^{D \times N}$ such that $\tilde{f}_i/\|\tilde{f}_i\| = \tilde{F}v_i$, and $\bar{g} = g(f(\lambda x_i + (1-\lambda)x_j))/\|g(f(\lambda x_i + (1-\lambda)x_j))\|$ for conciseness.

$$
\ell_{\text{BYOL}}^{i\text{-Mix}}\big((x_i, v_i), (x_j, v_j); \mathcal{B}, \lambda\big) = \ell_{\text{BYOL}}(\lambda x_i + (1 - \lambda)x_j, \lambda v_i + (1 - \lambda)v_j)
$$

$$
= \left\| \bar{g} - \tilde{F}(\lambda v_i + (1 - \lambda)v_j) \right\|^2 = \left\| \bar{g} - \left(\lambda \tilde{F}v_i + (1 - \lambda)\tilde{F}v_j\right) \right\|^2
$$

$$
= 1 - 2 \cdot \bar{g}^\top \left(\lambda \tilde{F}v_i + (1 - \lambda)\tilde{F}v_j\right) + \left\| \lambda \tilde{F}v_i + (1 - \lambda)\tilde{F}v_j \right\|^2
$$

$$
= 2 - 2 \cdot \bar{g}^\top \left(\lambda \tilde{F}v_i + (1 - \lambda)\tilde{F}v_j\right) + \text{const}
$$

$$
= \lambda \|\bar{g} - \tilde{F}v_i\|^2 + (1 - \lambda)\|\bar{g} - \tilde{F}v_j\|^2 + \text{const}
$$

$$
= \lambda \ell_{\text{BYOL}}(\lambda x_i + (1 - \lambda)x_j, v_i; \mathcal{B}) + (1 - \lambda)\ell_{\text{BYOL}}(\lambda x_i + (1 - \lambda)x_j, v_j; \mathcal{B}) + \text{const}. \tag{B.2}
$$

Because $\tilde{F}$ is not backpropagated, it can be considered as a constant.

# C    MORE ON EXPERIMENTS

We describe details of the experimental settings and more experimental results. For additional experiments below, we adapted the code for supervised contrastive learning (Khosla et al., 2020).[13]

## C.1    SETUP

In this section, we describe details of the experimental settings. Note that the learning rate is scaled by the batch size (Goyal et al., 2017): ScaledLearningRate = LearningRate × BatchSize/256.

**Image.** The experiments on CIFAR-10 and 100 (Krizhevsky & Hinton, 2009) and ImageNet (Deng et al., 2009) are conducted in two stages: following Chen et al. (2020a), the convolutional neural network (CNN) part of ResNet-50 (He et al., 2016)[14] followed by the two-layer multilayer perceptron (MLP) projection head (output dimensions are 2048 and 128, respectively) is trained on the unlabeled pretext dataset with a batch size of 256 (i.e., 512 augmented data) with the stochastic gradient descent (SGD) optimizer with a momentum of 0.9 over up to 4000 epochs. BYOL has an additional prediction head (output dimensions are the same with the projection head), which follows the projection head, only for the model updated by gradient. 10 epochs of warmup with a linear schedule to an initial learning rate of 0.125, followed by the cosine learning rate schedule (Loshchilov & Hutter, 2017) is used. We use the weight decay of 0.0001 for the first stage. For ImageNet, we use the same hyperparameters except that the batch size is 512 and the initial learning rate is 0.03.

---

[13] https://github.com/HobbitLong/SupContrast
[14] For small resolution data from CIFAR and Speech Commands, we replaced the kernal, stride, and padding size from (7,2,3) to (3,1,1) in the first convolutional layer, and removed the first max pooling layer, following Chen et al. (2020a).

Then, the head of the CNN is replaced with a linear classifier, and only the linear classifier is trained with the labeled downstream dataset. For the second stage, we use a batch size of 256 with the SGD optimizer with a momentum of 0.9 and an initial learning rate chosen among $\{1, 3, 5, 10, 30, 50, 70\}$ over 100 epochs, where the learning rate is decayed by 0.2 after 80, 90, 95 epochs. No weight decay is used at the second stage. The quality of representation is evaluated by the top-1 accuracy on the downstream task. We sample a single mixing coefficient $\lambda \sim \text{Beta}(1, 1)$ for each training batch. The temperature is set to $\tau = 0.2$. Note that the optimal distribution of $\lambda$ and the optimal value of $\tau$ varies over different architectures, methods, and datasets, but the choices above result in a reasonably good performance. The memory bank size of MoCo is 65536 for ImageNet and 4096 for other datasets, and the momentum for the exponential moving average (EMA) update is 0.999 for MoCo and BYOL. We do not symmetrize the BYOL loss, as it does not significantly improve the performance while increasing computational complexity.

For data augmentation, we follow Chen et al. (2020a): We apply a set of data augmentations randomly in sequence including resized cropping (Szegedy et al., 2015), horizontal flipping with a probability of 0.5, color jittering,[15] and gray scaling with a probability of 0.2. A Gaussian blurring with $\sigma \in [0.1, 2]$ and kernel size of 10% of the image height/width is applied for ImageNet. For evaluation on downstream tasks, we apply padded cropping with the pad size of 4 and horizontal flipping for CIFAR-10 and 100, and resized cropping and horizontal flipping for ImageNet.

**Speech.** In the experiments on Speech Commands (Warden, 2018), the network is the same with the image domain experiments, except that the number of input channels is one instead of three. The temperature is set to $\tau = 0.5$ for the standard setting and $\tau = 0.2$ for the no augmentation setting. 10% of silence data (all zero) are added when training. At the first stage, the model is trained with the SGD optimizer with a momentum of 0.9 and an initial learning rate of 0.125 over 500 epochs, where the learning rate decays by 0.1 after 300 and 400 epochs and the weight decay is 0.0001. The other settings are the same with the experiments on CIFAR.

For data augmentation,[16] we apply a set of data augmentations randomly in sequence including changing amplitude, speed, and pitch in time domain, stretching, time shifting, and adding background noise in frequency domain. Each data augmentation is applied with a probability of 0.5. Augmented data are then transformed to the mel spectogram in the size of $32 \times 32$.

**Tabular.** In the experiments on CovType and Higgs (Asuncion & Newman, 2007), we take a five-layer MLP with batch normalization as a backbone network. The output dimensions of layers are (2048-2048-4096-4096-8192), where all layers have batch normalization followed by ReLU except for the last layer. The last layer activation is maxout (Goodfellow et al., 2013) with 4 sets, such that the output dimension is 2048. On top of this five-layer MLP, we attach two-layer MLP (2048-128) as a projection head. We sample a single mixing coefficient $\lambda \sim \text{Beta}(\alpha, \alpha)$ for each training batch, where $\alpha = 2$ for CovType and Higgs100k, and $\alpha = 1$ for Higgs1M. The temperature is set to $\tau = 0.1$. The other settings are the same with the experiments on CIFAR, except that the batch size is 512 and the number of training epochs is 500. At the second stage, the MLP head is replaced with a linear classifier. For Higgs, the classifier is computed by linear regression from the feature matrix obtained without data augmentation to the label matrix using the pseudoinverse. Since the prior knowledge on tabular data is very limited, only the masking noise with a probability of 0.2 is considered as a data augmentation.

## C.2 VARIATIONS OF $i$-MIX

We compare the MixUp (Zhang et al., 2018) and CutMix (Yun et al., 2019) variation of $i$-Mix on N-pair contrastive learning and SimCLR. To distinguish them, we call them $i$-MixUp and $i$-CutMix, respectively. To be fair with the memory usage in the pretext task stage, we reduce the batch size of $i$-MixUp and $i$-CutMix by half (256 to 128) for SimCLR. Following the learning rate adjustment strategy in Goyal et al. (2017), we also decrease the learning rate by half (0.125 to 0.0625) when the batch size is reduced. We note that $i$-MixUp and $i$-CutMix on SimCLR take approximately 2.5 times more training time to achieve the same number of training epochs. The results are provided in Table C.1. We first verify that the N-pair formulation results in no worse performance than that of SimCLR. This justifies to conduct experiments using the N-pair formulation instead of that of

---

[15]Specifically, brightness, contrast, and saturation are scaled by a factor uniformly sampled from $[0.6, 1.4]$ at random, and hue is rotated in the HSV space by a factor uniformly sampled from $[-0.1, 0.1]$ at random.

[16]https://github.com/tugstugi/pytorch-speech-commands

| Pretext | Downstream | N-pair | | | SimCLR | | |
|---|---|---|---|---|---|---|---|
| | | Vanilla | $i$-MixUp | $i$-CutMix | Vanilla | $i$-MixUp | $i$-CutMix |
| CIFAR-10 | CIFAR-10 | 92.4 $\pm$ 0.1 | **94.8** $\pm$ 0.2 | 94.7 $\pm$ 0.1 | 92.5 $\pm$ 0.1 | **94.8** $\pm$ 0.2 | **94.8** $\pm$ 0.2 |
| | CIFAR-100 | 60.2 $\pm$ 0.3 | **63.3** $\pm$ 0.2 | 61.5 $\pm$ 0.2 | 60.0 $\pm$ 0.2 | **61.4** $\pm$ 1.0 | 57.1 $\pm$ 0.4 |
| CIFAR-100 | CIFAR-10 | 84.4 $\pm$ 0.2 | **86.2** $\pm$ 0.2 | 85.1 $\pm$ 0.2 | 84.4 $\pm$ 0.2 | **85.2** $\pm$ 0.3 | 83.7 $\pm$ 0.6 |
| | CIFAR-100 | 68.7 $\pm$ 0.2 | **72.3** $\pm$ 0.2 | **72.3** $\pm$ 0.4 | 68.7 $\pm$ 0.2 | **72.3** $\pm$ 0.2 | 71.7 $\pm$ 0.2 |

Table C.1: Comparison of N-pair contrastive learning and SimCLR with $i$-MixUp and $i$-CutMix on them with ResNet-50 on CIFAR-10 and 100. We run all experiments for 1000 epochs. $i$-MixUp improves the accuracy on the downstream task regardless of the data distribution shift between the pretext and downstream tasks. $i$-CutMix shows a comparable performance with $i$-MixUp when the pretext and downstream datasets are the same, but it does not when the data distribution shift occurs.

| Pretext | Downstream | Self-Supervised Pretext | | | Supervised Pretext | | |
|---|---|---|---|---|---|---|---|
| | | SimCLR | N-pair | + $i$-Mix | SimCLR | N-pair | + $i$-Mix |
| CIFAR-10 | CIFAR-10 | 92.5 $\pm$ 0.1 | 92.4 $\pm$ 0.1 | **94.8** $\pm$ 0.2 | 95.6 $\pm$ 0.3 | 95.7 $\pm$ 0.1 | **97.0** $\pm$ 0.1 |
| | CIFAR-100 | 60.0 $\pm$ 0.2 | 60.2 $\pm$ 0.3 | **63.3** $\pm$ 0.2 | 58.6 $\pm$ 0.2 | **58.9** $\pm$ 0.5 | 57.8 $\pm$ 0.6 |
| CIFAR-100 | CIFAR-10 | 84.4 $\pm$ 0.2 | 84.4 $\pm$ 0.2 | **86.2** $\pm$ 0.2 | 86.5 $\pm$ 0.4 | 86.7 $\pm$ 0.2 | **88.7** $\pm$ 0.2 |
| | CIFAR-100 | 68.7 $\pm$ 0.2 | 68.7 $\pm$ 0.2 | **72.3** $\pm$ 0.2 | 74.3 $\pm$ 0.2 | 74.6 $\pm$ 0.3 | **78.4** $\pm$ 0.2 |

Table C.2: Comparison of the N-pair self-supervised and supervised contrastive learning methods and $i$-Mix on them with ResNet-50 on CIFAR-10 and 100. We also provide the performance of formulations proposed in prior works: SimCLR (Chen et al., 2020a) and its supervised version (Khosla et al., 2020). We run all experiments for 1000 epochs. $i$-Mix improves the accuracy on the downstream task regardless of the data distribution shift between the pretext and downstream tasks, except the case that the pretest task has smaller number of classes than that of the downstream task. The quality of representation depends on the pretext task in terms of the performance of transfer learning: self-supervised learning is better on CIFAR-10, while supervised learning is better on CIFAR-100.

SimCLR, which is simpler and more efficient, especially when applying $i$-Mix, while not losing the performance. When pretext and downstream tasks share the training dataset, $i$-CutMix often outperforms $i$-MixUp, though the margin is small. However, $i$-CutMix shows a worse performance in transfer learning.

Table C.2 compares the performance of SimCLR, N-pair contrastive learning, and $i$-Mix on N-pair contrastive learning when the pretext task is self-supervised and supervised contrastive learning. We confirm that the N-pair formulation results in no worse performance than that of SimCLR in supervised contrastive learning as well. $i$-Mix improves the performance of supervised contrastive learning from 95.7% to 97.0% on CIFAR-10, similarly to improvement achieved by MixUp for supervised learning where it improves the performance of supervised classifier learning from 95.5% to 96.6%. On the other hand, when the pretext dataset is CIFAR-100, the performance of supervised contrastive learning is not better than that of supervised learning: MixUp improves the performance of supervised classifier learning from 78.9% to 82.2%, and $i$-Mix improves the performance of supervised contrastive learning from 74.6% to 78.4%.

While supervised $i$-Mix improves the classification accuracy on CIFAR-10 when trained on CIFAR-10, the representation does not transfer well to CIFAR-100, possibly due to overfitting to 10 class classification. When pretext dataset is CIFAR-100, supervised contrastive learning shows a better performance than self-supervised contrastive learning regardless of the distribution shift, as it learns sufficiently general representation for linear classifier to work well on CIFAR-10 as well.

### C.3 QUALITATIVE EMBEDDING ANALYSIS

Figure C.1 visualizes embedding spaces learned by N-pair contrastive learning and $i$-Mix on CIFAR-10 and 100. When the downstream dataset is the same with the pretext task, both contrastive learning and $i$-Mix cluster classes well, as shown in Figure C.1(a) and C.1(b). However, when the downstream task is transferred to CIFAR-100, $i$-Mix in Figure C.1(d) clusters classes better than contrastive

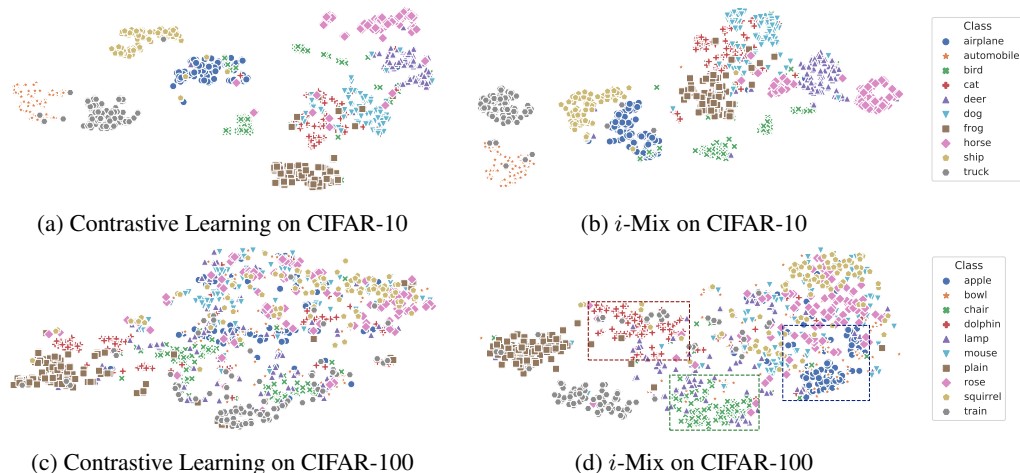

(a) Contrastive Learning on CIFAR-10      (b) $i$-Mix on CIFAR-10

(c) Contrastive Learning on CIFAR-100      (d) $i$-Mix on CIFAR-100

Figure C.1: t-SNE visualization of embeddings trained by contrastive learning and $i$-Mix with ResNet-50 on CIFAR-10. (a,b): Classes are well-clustered in both cases when applied to CIFAR-10. (c,d): When models are transferred to CIFAR-100, classes are more clustered for $i$-Mix than contrastive learning, as highlighted in dashed boxes. We show 10 classes for a better visualization.

| Pretext | Downstream | FED ($\times 10^{-4}$) ($\downarrow$) | | Training Acc (%) ($\uparrow$) | | Test Acc (%) ($\uparrow$) | |
|---|---|---|---|---|---|---|---|
| | | N-pair | + $i$-Mix | N-pair | + $i$-Mix | N-pair | + $i$-Mix |
| CIFAR-10 | CIFAR-10 | 30.0 | **16.7** | **96.1** | **96.1** | 92.4 | **94.8** |
| | CIFAR-100 | 13.8 | **7.9** | **70.7** | 69.5 | 60.2 | **63.3** |
| CIFAR-100 | CIFAR-10 | 15.2 | **9.7** | 88.1 | **88.8** | 84.4 | **86.2** |
| | CIFAR-100 | 30.4 | **13.3** | **85.6** | 79.0 | 68.7 | **72.3** |

Table C.3: Comparison of N-pair contrastive learning and $i$-Mix with ResNet-50 on CIFAR-10 and 100 in terms of the Fréchet embedding distance (FED) between training and test data distribution on the embedding space, and training and test accuracy. $\uparrow$ ($\downarrow$) indicates that the higher (lower) number is the better. $i$-Mix improves contrastive learning in all metrics, which shows that $i$-Mix is an effective regularization method for the pretext task, such that the learned representation is more generalized.

learning in Figure C.1(c). Specifically, clusters of "apple," "chair," and "dolphin," can be found in Figure C.1(d) while they spread out in Figure C.1(c). Also, "rose" and "squirrel" are more separated in Figure C.1(d) than C.1(c). This shows that the representation learned with $i$-Mix is more generalizable than vanilla contrastive learning.

## C.4   QUANTITATIVE EMBEDDING ANALYSIS

To estimate the quality of representation by the similarity between training and test data distribution, we measure the Fréchet embedding distance (FED): similarly to the Fréchet inception distance (FID) introduced in Heusel et al. (2017), FED is the Fréchet distance (Fréchet, 1957; Vaserstein, 1969) between the set of training and test embedding vectors under the Gaussian distribution assumption. For conciseness, let $\bar{f}_i = f(x_i)/\|f(x_i)\|$ be an $\ell_2$ normalized embedding vector; we normalize embedding vectors as we do when we measure the cosine similarity. Then, with the estimated mean $m = \frac{1}{N}\sum_{i=1}^{N}\bar{f}_i$ and the estimated covariance $S = \frac{1}{N}\sum_{i=1}^{N}(\bar{f}_i - m)(\bar{f}_i - m)^\top$, the FED can be defined as

$$d^2\big((m^{\text{tr}}, S^{\text{tr}}), (m^{\text{te}}, S^{\text{te}})\big) = \|m^{\text{tr}} - m^{\text{te}}\|^2 + \text{Tr}\big(S^{\text{tr}} + S^{\text{te}} - 2(S^{\text{tr}}S^{\text{te}})^{\frac{1}{2}}\big). \quad \text{(C.1)}$$

As shown in Table C.3, $i$-Mix improves FED over contrastive learning, regardless of the distribution shift. Note that the distance is large when the training dataset of the downstream task is the same with that of the pretext task. This is because the model is overfit to the training dataset, such that the distance from the test dataset, which is unseen during training, has to be large.

On the other hand, Table C.3 shows that $i$-Mix reduces the gap between the training and test accuracy. This implies that $i$-Mix is an effective regularization method for pretext tasks, such that the learned representation is more generalizable on downstream tasks.

