# OpenReview forum: "$i$-Mix: A Domain-Agnostic Strategy for Contrastive Representation Learning"
_ICLR.cc/2021/Conference — ICLR 2021 Poster_

### Official Review · AnonReviewer2 · 2020-10-19
**Mixup + contrastive learning yields great results**

**Rating:** 7
**Confidence:** 5

**Review:**

Summary: The authors present a way of framing contrastive self-supervised learning techniques that enables them to use MixUp. Results indicate this helps especially on small datasets, and in domains where the right set of data augmentations is unknown.

Pros:
+ The proposed approach for adapting the effective Mix-Up strategy to contrastive learning is simple. At the same time it is elegant in its generality: it improves all self-supervised learning techniques. There have been very few such general-purpose innovations in self-supervised learning since the original instance discrimination work that started the current trend.
+ The fact that iMix helps a lot when domain-specific data augmentations are unknown is a very significant result, since it opens up self-supervised learning to a lot of new domains.
+ I find the experimental evaluation quite careful and detailed. I especially like that the authors went beyond images as a domain; the self-supervised learning community has been overfitting to imagenet.

Cons: TBH I can't think of any. This clearly seems to be a worthwhile extension to the world of self-supervised techniques, especially relevant for new domains with limited data.

---

> ### Author Response · Authors · 2020-11-19
> **Response to AnonReviewer2**
>
> We appreciate your valuable comments. As you said, $i$-Mix can be considered as a general-purpose method improving the performance of self-supervised learning algorithms, including SimCLR, MoCo, and BYOL. Our experimental results in multiple domains support it.
>
> We've updated the manuscript following reviewers' suggestions. Please take a look and don't hesitate to remind us if we have missed something or there are more suggestions.

---

### Official Review · AnonReviewer4 · 2020-10-27
**Strong empirical results, but a few clarifications are needed.**

**Rating:** 7
**Confidence:** 5

**Review:**

In essence, the premise and contribution of the paper is rather simple -- contrastive learning and mixup are both very effective methods in their respective domains, and this paper proposes the use of mixup to improve contrastive learning, and the results seem to indicate this.

Positives:
- Strong empirical results, validated on a rather wide range of settings, including dataset type (image, audio, or tabular) as well as various contrastive techniques (SimCLR, BYOL, MoCO), and ablations (dataset size and model capacity).

Suggestions / negatives:

- (1) I see that you experimented with CutMix [3] and that you reported some results in the supplementary material. Can you comment further on this? (Also, see (3)). I am also concerned with the potential of vanilla mixup to 'underfit' (see [1]), though perhaps this would be mitigated via careful tuning of the mixing distribution parameters 'alpha' (again, see (3)).
- (2) Please include measures of uncertainty in your results. This is lacking in all of your tables. You mention the results are an average of up to 5 runs, but the variance should still be stated.
- (3) Did you do any HP tuning on the alpha for Beta(alpha, alpha)? Section 4 intro says that it was fixed to 1.0 for image datasets and 2.0 for tabular. Was this the result of HP tuning? And if so, can you comment on if the choice of alpha made a big difference?

Extra suggestions / comments:
- (4) It would be nice to test the effect of mixup without modifying the contrastive learning algorithm, that way you can examine its effects purely as a data augmentation scheme. It turns out that if you use Beta(alpha, alpha+1) as your mixing distribution, it is equivalent in expectation to the mixup loss where labels are *not* mixed. This applies to the categorical cross-entropy and binary cross-entropy loss functions. See [2] for the proof of this. This means that you could simply take the algorithm you have already, make the sampling distribution Beta(alpha, alpha+1) (rather than Beta(alpha,alpha)), and then simply use CCE(L, Y) as the loss, where you are *not* mixing Y's. I'm curious to know how this works, since you would be able to test mixup in isolation without having to modify the contrastive algorithm itself. Also, note that this particular parameterisation of the mixing distribution is Beta(alpha,alpha+1) rather than Beta(alpha, alpha), and maybe it is worth seeing if it either one performs better than the other.

In summary, I think what you have here is good, but you should add measures of uncertainty to your results and also clarify how you performed the hyperparameter tuning. I'd be very interested if you address (4) as well. My rating is a 6, though I open to raising the score if my concerns have been addressed.

References:

- [1]: Guo, H., Mao, Y., & Zhang, R. (2019, July). Mixup as locally linear out-of-manifold regularization. In Proceedings of the AAAI Conference on Artificial Intelligence (Vol. 33, pp. 3714-3722).
- [2]: Huszár, F. (2017). Mixup: data-dependent data augmentation. Blog post: https://www.inference.vc/mixup-data-dependent-data-augmentation/
- [3]: Yun, S., Han, D., Oh, S. J., Chun, S., Choe, J., & Yoo, Y. (2019). Cutmix: Regularization strategy to train strong classifiers with localizable features. In Proceedings of the IEEE International Conference on Computer Vision (pp. 6023-6032).

---

> ### Author Response · Authors · 2020-11-19
> **Response to AnonReviewer4**
>
> We appreciate your valuable comments. We've updated the manuscript following reviewers' suggestions. Please take a look and don't hesitate to remind us if we have missed something or there are more suggestions.
>
> (1) CutMix
>
> As our focus is not on comparing MixUp variants, we put the comparison between the MixUp and CutMix version of $i$-Mix on CIFAR datasets in Table C.1 in Appendix. $i$-CutMix performs comparably with $i$-MixUp when the pretext and downstream datasets are the same, but it does not when two datasets are different.
>
> (2) Statistics of experimental results
>
> As reported in the first paragraph in Section 4, all numbers in our main experimental results are the average of 3--5 runs except for those of large datasets. We note that most experiments have a small standard deviation (less than 1%; mostly 0.1--0.3%), while the performance gain by $i$-Mix is significant in all small datasets; for example, 6.1% when $i$-Mix is applied to MoCo on CIFAR-100. As suggested by R1 and R4, we put the standard deviation to tables in the updated manuscript.
>
> Regarding the standard deviation of large datasets, please note that prior contrastive representation learning works including [[MoCo](arxiv.org/pdf/1911.05722.pdf)], [[MoCo v2](arxiv.org/pdf/2003.04297.pdf)], [[SimCLR](arxiv.org/pdf/2002.05709.pdf)], and [[BYOL](arxiv.org/pdf/2006.07733.pdf)] also did not report the standard deviation of performances, where they mostly experimented on large image datasets. Please note that experiments on large datasets require a lot of resources (e.g., training MoCo v2 with ResNet-50 on ImageNet takes 9 days with 8 V100 GPUs, or around 1 month with 8 Titan X GPUs). Fortunately, the linear classification performance seems not varying much, as we could replicate the reported performance of MoCo v2.
>
> (3) Hyperparameter tuning for $\alpha$ and underfitting
>
> Thanks for referring to [[1]](arxiv.org/pdf/1809.02499.pdf). Different from supervised learning results in [1], $i$-Mix on contrastive representation learning shows generally a good performance for $\alpha \in [0.2, 2]$ in our experiments. We did not tune $\alpha$ in a high resolution, but found that $\alpha=1$ is generally good for most cases. As suggested, tuning $\alpha$ more carefully or applying the idea of [1] would result in a better performance in practice.
>
> We used $\alpha=2$ for tabular datasets for stronger regularization as we have a limited number of data augmentations in this domain.
>
> (4) InputMix with $\text{Beta}(\alpha, \alpha+1)$
>
> Thanks for sharing [[2]](https://www.inference.vc/mixup-data-dependent-data-augmentation/), but the derivation might be flawed. Specifically, changing $\lambda$ to a Bernoulli variable partially for $y$ would be problematic.
>
> We also empirically confirmed that InputMix with $\text{Beta}(\alpha, \alpha+1)$ and $i$-Mix with $\text{Beta}(\alpha, \alpha)$ have different loss values:
> when `input = torch.eye(100, 100)*10 + torch.randn(100,100)` and no model is used (so input is directly used as output or logit), the average loss of contrastive learning, $i$-Mix with $\text{Beta}(\alpha, \alpha)$, and InputMix with $\text{Beta}(\alpha, \alpha+1)$ over 100 trials is 0.0141, 1.0606, and 0.2482, respectively. For replicating these results, please run `python toy_inputmix.py` in the code repository we shared in another comment.
>
> In our preliminary experiments, InputMix as a standalone method was not better than $i$-Mix in general. However, we think the derivation above is still interesting, so we experimented it and share the MoCo v2 results trained for 1000 epochs on CIFAR-100:
>
> | Method | Beta | Top 1 (%) |
> | :-: | :-: | :-: |
> | MoCo v2 | - | 71.7 |
> | InputMix | $\text{Beta}(1,1)$ | 73.3 |
> | InputMix | $\text{Beta}(1,2)$ [[2]](https://www.inference.vc/mixup-data-dependent-data-augmentation/) | 72.5 |
> | $i$-Mix | $\text{Beta}(1,1)$ | 77.7 |
>
> In the code we shared in another comment, you can try this by adding following arguments:
> InputMix with $\text{Beta}(1,1)$: `--inputmix 1 --alpha2 1.0`
> InputMix with $\text{Beta}(1,2)$: `--inputmix 1 --alpha2 1.0 --ber`

---

> > ### Comment · AnonReviewer4 · 2020-11-20
> > **Thanks**
> >
> > Thanks for the clarification.
> >
> > **Regarding the standard deviation of large datasets, please note that prior contrastive representation learning works including [MoCo], [MoCo v2], [SimCLR], and [BYOL] also did not report the standard deviation of performances, where they mostly experimented on large image datasets.** Yes, it is very much the case that many papers in our field completely neglect statistics, but we should always lead by example.

---

### Official Review · AnonReviewer3 · 2020-10-28
**Timely topic, simple and effective method, would benefit from a more few baselines**

**Rating:** 7
**Confidence:** 3

**Review:**

Summary:

The key idea of this paper is to apply MixUp-style regularization to self-supervised contrastive learning techniques (SimCLR, MoCo-v2, BYOL). This is combined with another form of MixUp that involves only the images (not the labels), but the precise nature of this component is unclear. For large networks trained on small datasets, the proposed method improves downstream classification performance by reducing overfitting (Table 1, Figure 2). The results are mixed for large-scale datasets (Table 2, Figure 3). The proposed method is also investigated as a potential domain-agnostic augmentation. Though much better than not using any augmentations at all, the results are not typically better than using standard augmentations (Table 3).

Strengths:

The paper is generally clear and the figures are well-made, though the presentation could be improved here and there (see minor comments below). The question of regularizing self-supervised representation learning is an interesting one - most contrastive learning papers use such large datasets that overfitting isn't really an issue, so that topic is novel as far as I know. The proposed method is simple and effective, clearly improving the performance of several contrastive learning techniques in the regime where overfitting is an issue.

Weaknesses:

The paper proposes a regularization method for contrastive learning and demonstrates that it can help to reduce overfitting. However, it has not been demonstrated that the proposed method is unusual in this regard. Perhaps simpler, more traditional regularization methods (e.g. dropout, weight decay) would have the same effect.

As far as I can tell, the proposed method actually has two components - i-Mix and InputMix - whose effects have not been separated by an ablation study.

Regarding the observation that "i-Mix + linear classifier" can outperform supervised baselines (Section 4.2) - it would be interesting to know whether this is still true when those supervised baselines also get to benefit from a regularization technique like MixUp. It seems like that might be a more fair comparison.

Overall:

The role of overfitting in contrastive self-supervised learning is an interesting and timely topic, and the proposed method is simple, effective, and general. However, additional baselines would make the importance of this particular regularization technique more apparent.

Minor comments:

Section 3.2: Perhaps $v_i \in [0,1]^N$ should be $v_i \in \{0,1\}^N$.

Section 3.3:

"...we propose to apply InputMix together with i-Mix, which mixes input data but not their labels" - this a somewhat confusing phrase, since the description could be read as applying to either InputMix or i-Mix.

What does "with the largest mixing coefficient $\lambda$" mean? Do we sample $\lambda$ first, then sample another mixing coefficient on $[0, \lambda]$?

This section is the only one where InputMix is mentioned - are we to assume that it is used in all cases where i-Mix is used?

Section 4.0: It might be helpful to point out that $\mathrm{Beta}(1,1)$ is the uniform distribution (i.e. the standard setting for MixUp). Is there any intuition behind changing the distribution of $\lambda$ for tabular data?

Section 4.1:

Are the 32x32 images resized for input to the network? Or is the network modified at all for the small images?

Section 4.3:

It would be nice to see experiments of this kind on ImageNet, which would allow the enormous models and longer training times to be balanced out by a suitably sized pretraining dataset.

In Figure 3, the caption says "MoCo" but the legend says "N-pair" - not sure which it should be. Also, consider including "10% of ImageNet" information in the caption.

Section 4.3: How is this 10% ImageNet dataset constructed?

Section 4.4:

It would also be helpful to explicitly compare the "no augmentation + i-Mix" performance against supervised numbers so the reader can easily tell whether i-Mix by itself results in a strong pretraining strategy or not.

All sections:

I recommend replacing $\beta(\alpha, \alpha)$ with $\mathrm{Beta}(\alpha, \alpha)$ wherever it occurs in the paper.

The experiments can be a bit "patchy" - different self-supervised methods used for different experiments on different datasets. If possible, consider fleshing out e.g. Table 2 to cover more of the dataset/method combinations.

I would write "MoCov2" instead of "MoCo" so people don't get the wrong impression if they're just glancing through the paper.

**Update: As noted elsewhere in the discussion, the authors have addressed my primary concern. I will therefore increase my score from a 6 to a 7.**

---

> ### Author Response · Authors · 2020-11-19
> **Response to AnonReviewer3 (1/2)**
>
> We appreciate your valuable comments. We've updated the manuscript following reviewers' suggestions. Please take a look and don't hesitate to remind us if we have missed something, or there are more suggestions.
>
> (1) Other traditional regularization methods as baselines: Dropout
>
> Dropout is known to be often incompatible with BN, while BN is an essential component in our baselines, ([[especially in BYOL](https://untitled-ai.github.io/understanding-self-supervised-contrastive-learning.html)]), so applying both dropout and BN would be another challenge [[Li et al., 2019](https://openaccess.thecvf.com/content_CVPR_2019/papers/Li_Understanding_the_Disharmony_Between_Dropout_and_Batch_Normalization_by_Variance_CVPR_2019_paper.pdf)]. In our very initial experiments, we've tried adding dropout to contrastive representation learning models, but the performance was not better. Below we share the results when we apply dropout with a droprate of 0.2 to MoCo v2 trained for 1000 epochs on CIFAR100.
>
> | Method | Drop Rate | Top 1 (%) |
> | :-: | :-: | :-: |
> | MoCo v2 | 0 | 71.7 |
> | + $i$-Mix | 0 | **77.7** |
> | MoCo v2 | 0.2 | 55.8 |
> | + $i$-Mix | 0.2 | **62.1** |
>
> (2) Other traditional regularization methods as baselines: Weight decay
>
> We agree that tuning the weight decay is important. A proper weight decay is already applied in our experiments. Below we share our ablation study results w.r.t. weight decay on MoCo v2 trained for 1000 epochs on CIFAR100. While the performance of MoCo v2 is sensitive to weight decay, applying $i$-Mix consistently improves the performance.
>
> | Method | Weight Decay | Top 1 (%) |
> | :-: | :-: | :-: |
> | MoCo v2 | 1e-3 | 68.3 |
> | + $i$-Mix | 1e-3 | **70.2** |
> | MoCo v2 | 1e-4 | 71.7 |
> | + $i$-Mix | 1e-4 | **77.7** |
> | MoCo v2 | 1e-5 | 65.1 |
> | + $i$-Mix | 1e-5 | **73.2** |
>
> (3) InputMix
>
> We clarify that InputMix is applied "together with i-Mix" on image datasets **only in Table 3**. In other words, only $i$-Mix on CIFAR in Table 3 is $i$-Mix + InputMix, and InputMix is not used anywhere else.
>
> "with the largest mixing coefficient $\lambda$" means that the principal data takes the largest weight in the mixing operation. In experiments, we mixed one principal data and two auxiliary data for InputMix. For each index, ($0.5 \lambda_1 + 0.5$, $0.5 \lambda_2$, $0.5 \lambda_3$) are used as mixing coefficients, where $\lambda_1, \lambda_2, \lambda_3 \sim \text{Dir}(1,1,1)$. We added this detailed description in Section 4.4.
> The effect of InputMix on image datasets in Table 3 is significant:
>
> | Dataset | N-pair | $i$-Mix | $i$-Mix + InputMix |
> | :-: | :-: | :-: | :-: |
> | CIFAR-10 | 17.0 | 49.6 | 79.7 |
> | CIFAR-100 | 3.0 | 24.3 | 50.7 |
>
> However, we couldn't see a significant improvement in other cases, especially when other data augmentations are applied.
>
> (4) Comparison between $i$-Mix + linear classifier and supervised learning (+ MixUp) in Section 4.2 and 4.4
>
> Thanks for the suggestion. For the comparison with supervised learning in Section 4.2, we followed the convention ([[SimCLR](arxiv.org/pdf/2002.05709.pdf)], [[BYOL](arxiv.org/pdf/2006.07733.pdf)]), which employs the simplest end-to-end supervised learning model. While this comparison is not meant to be apples to apples, as we are not allowed to use labels during self-supervised learning, we agree that it could be misleading to claim our performance is as good as a fully supervised model. We clarified this in a footnote in Section 4.2.
>
> On the other hand, since the performance of self-supervised representation learning relies more on strong data augmentations than supervised learning, the performance gap between them will be larger in the no-augmentation setting in Section 4.4, i.e., the supervised learning performance is better than self-supervised learning in classification tasks.
>
> (5) Beta distribution in tabular datasets
>
> We used $\alpha=2$ for tabular datasets for stronger regularization as we have a limited number of data augmentations in this domain. We added a brief description about the shape of Beta distribution with respect to $\alpha$ in a footnote.
>
> (6) Network for 32x32 images
>
> Following Appendix B.9 in [[SimCLR](arxiv.org/pdf/2002.05709.pdf)], we changed the first conv layer (more specifically, (kernel size, stride, padding) is changed from (7,2,3) to (3,1,1)) and removed the first maxpool. We added this detailed description in a footnote in Appendix C.1.

---

> > ### Author Response · Authors · 2020-11-19
> > **Response to AnonReviewer3 (2/2)**
> >
> > (7) Controlled experiments on ImageNet in Section 4.3
> >
> > The focus of our paper is mostly on small-scale settings where $i$-Mix shows a significant gain, and unfortunately we do not have enough resources to scale up all controlled experiments on ImageNet. Instead, we have been conducting more experiments on a subset of ImageNet (e.g., 10%) to support our claim. For example, the updated Table 2 shows that when a subset of ImageNet with 100 classes is used for both pretext and downstream tasks, $i$-Mix improves the MoCo v2 baseline in a larger margin than when 1k classes are used, from 84.1% to 87.0%.
> >
> > (8) Legend and caption in Figure 3: Thanks for catching the typo in the Figure 3 legend, the baseline is MoCo v2. We also replaced the figure with linear classification results on the full ImageNet.
> >
> > (9) 10% ImageNet is randomly sampled with different random seeds per class, but the sampled subset is the same for all experiments for a fair comparison.
> >
> > (10) The experiments can be a bit "patchy"
> >
> > Table 1 indeed covers different combinations of self-supervised learning methods and datasets. In the rest of experiments, we used N-pair contrastive learning for all datasets except MoCo for ImageNet, because SimCLR or N-pair contrastive learning on ImageNet requires a lot of GPUs in parallel for maintaining a large batch size during training.

---

> > ### Comment · AnonReviewer4 · 2020-11-20
> > **Table 3**
> >
> > Hi,
> >
> > I think to minimise confusion, you should rework Table 3 so that it has the "n-pair" followed by "i-mix" followed by "i-mix + inputmix", as you've stated in your above reply. It seems like you do have a bit of empty space left in the document to fill a full 9 pages so I'd recommend that. You could also state in your table caption that what you're trying to do here is test the effect of InputMix in a hypothetical scenario where we see what happens when we don't use domain-specific data augmentations.
> >
> > The use of Dirichlet mixup is interesting here, it would be worth citing [1] for this. Section 3.3 could mention that you are performing that mix specifically by doing $(0.5\lambda_1 + 0.5, 0.5\lambda_2, 0.5\lambda_3)$, otherwise if you have no room to add it, just refer the reader to the appendix.
> >
> > Very minor note: there may be some Dirichlet distn. identities that say that sampling $\lambda_1, \lambda_2, \lambda_3$ and computing the linear combinations $(0.5\lambda_1 + 0.5, 0.5\lambda_2, 0.5\lambda_3)$ is equivalent to sampling from some new Dirichlet, but I haven't found any. Anyway, that is just more of a presentational remark.
> >
> > [1]: Beckham, C., Honari, S., Verma, V., Lamb, A. M., Ghadiri, F., Hjelm, R. D., ... & Pal, C. (2019). On adversarial mixup resynthesis. In Advances in neural information processing systems (pp. 4346-4357).

---

> > > ### Author Response · Authors · 2020-11-21
> > > **Table 3 and Section 4.4**
> > >
> > > (1) $i$-Mix and InputMix
> > >
> > > Thanks for the suggestion! To avoid any confusion around InputMix, we added a footnote and marks to clarify in which methods InputMix is applied within Table 3, and we added the comparison between $i$-Mix and $i$-Mix + InputMix in Section 4.4.
> > > Please note that our intention in Table 3 is to show that this kind of data-driven augmentation strategy is effective when other data augmentations are not available, so we compare $i$-Mix and $i$-Mix + InputMix in a different place.
> > >
> > > (2) Dirichlet distribution for InputMix
> > >
> > > Thanks for referring to [1], which sampled weights from the Dirichlet distribution for (N>2)-tuple MixUp for training autoencoders. We cited this work around the description about InputMix hyperparameters. Following your suggestion, we moved the detailed description about InputMix hyperparameters from Appendix to Section 4.4.
> > >
> > > For InputMix, we want to guarantee that the weight for principal data ($\lambda_1$) is always large enough, because we take the label of the principal data for training. By definition, the pdf of Dirichlet distribution has a non-zero value over a continuous space (0,1), such that it cannot guarantee a minimum probability for a certain $\lambda$. For example, $(\lambda_1, \lambda_2, \lambda_3) \sim \text{Dirichlet}(10,1,1)$ generally assigns a high probability to $\lambda_1$, but it still have a chance to assign a very low probability to $\lambda_1$.

---

### Official Review · AnonReviewer1 · 2020-10-29
**Nice problem to tackle, but questionable results**

**Rating:** 3
**Confidence:** 4

**Review:**

# Introduction
First off, thank you (authors) for taking the time to put this together.

The problem the authors are trying to tackle is meaningful (+1) and relevant. Contrastive learning is in dire need of removing the dependency of using transforms. This paper goes one step and even tries to generalize it across domains which is desperately needed in the field.

## Paper summary
The authors have taken an idea from supervised learning (MixUp) and derived the matching application for contrastive learning in hopes of removing the need for domain-specific augmentations.

They show how to apply their i-Mix to SimCLR, Moco and BYOL.

Finally, authors show results for vision, tabular and speech data using standard datasets for each.

Results have a few sections:
1. Evaluate cifar-10, cifar-100, Commands and CovType using Moco and BYOL in addition to i-Mix.
2. Evaluate larger datasets, imagenet and Higgs.
3. Evaluate how depth of a resnet and length of training are affected with i-Mix.
4. They show ablations with models trained without their standard data augmentations and i-Mix.
5. They evaluate transfer learning on vision (cifar-10 <=> cifar-100) and (Imagenet -> VOC detection)

## Strong points.

1. The overall goal they are looking at is impactful and critical.
2. They show results on small and large datasets.
3. They are attempting 3 domains with the same method.
4. Their most promising result is Table 3 which shows a nice gap between models trained without data augmentation but with i-Mix instead.

## Weak points
Their general results are not significant enough. Most results look like this (82.5 vs 82.7).
This hints at these results being achieved largely via hyperparameter tuning, instead of meaningfully providing an advantage.

## Suggestions for improvement
1. Show distributions for results instead of a single number. (ie: not 82.5 vs 82.7, but instead run 50-100 times and plot the histogram)
2. Many of the ideas mentioned here are also mentioned in non-cited relevant work (for example: [YADIM](https://arxiv.org/abs/2009.00104)).
3. Provide code. Since some of these claims come down to implementation details, it's important to see the code as well.

## Recommendation
Reject

---

> ### Author Response · Authors · 2020-11-19
> **Response to AnonReviewer1**
>
> We appreciate your valuable comments. We've updated the manuscript following reviewers' suggestions. Please take a look and don't hesitate to remind us if we have missed something, or there are more suggestions.
>
> (1) Statistics of experimental results
>
> As reported in the first paragraph in Section 4, all numbers in our main experimental results are the average of 3--5 runs except for those of large datasets. We note that most experiments have a small standard deviation (less than 1%; mostly 0.1--0.3%), while the performance gain by $i$-Mix is significant in all small datasets; for example, 6.1% when $i$-Mix is applied to MoCo on CIFAR-100. As suggested by R1 and R4, we put the standard deviation to tables in the updated manuscript.
>
> Here we also provide t-test results to show that the improvement is significant. Among image dataset results in Table 1, BYOL on CIFAR-10 has the smallest performance gain, and the t-value is 13.7. MoCo on CIFAR-100 has the largest performance gain, and the t-value is 23.3. Their corresponding p-values are very small (< 0.01), implying that the performance gain is significant in both cases.
>
> The example given by R1 is the VOC object detection performance in $AP_{50}$, where $i$-Mix shows a marginal improvement, because it is pre-trained on a large dataset; we discussed this trend in Section 4.3. We averaged the performance of 5 runs as described in [[MoCo v2](arxiv.org/pdf/2003.04297.pdf)], and the standard deviation of $AP_{50}$ is 0.2 for both MoCo v2 and $i$-Mix.
>
> Regarding the standard deviation of large datasets, please note that prior contrastive representation learning works including [[MoCo](arxiv.org/pdf/1911.05722.pdf)], [[MoCo v2](arxiv.org/pdf/2003.04297.pdf)], [[SimCLR](arxiv.org/pdf/2002.05709.pdf)], [[BYOL](arxiv.org/pdf/2006.07733.pdf)], and even [[YADIM](arxiv.org/pdf/2009.00104.pdf)] mentioned by R1 also did not report the standard deviation of performances, where they mostly experimented on large image datasets. Please note that experiments on large datasets require a lot of resources (e.g., training MoCo v2 with ResNet-50 on ImageNet takes 9 days with 8 V100 GPUs, or around 1 month with 8 Titan X GPUs). Fortunately, the linear classification performance seems not varying much, as we could replicate the reported performance of MoCo v2.
>
> Regarding hyperparameter tuning, we basically tuned the hyperparameters for respective baseline self-supervised learning algorithms, and applied the same hyperparameters for $i$-Mix. In fact, except for the number of epochs, hyperparameters are mostly unchanged from MoCo v2; we couldn't see a significant improvement by tuning hyperparameters further in both baselines and $i$-Mix.
>
> The only additional hyperparameter introduced by $i$-Mix is $\alpha$ to determine the shape of the beta distribution. We used $\alpha=1$ for most experiments as it is good enough to show the improvement by $i$-Mix. Tuning $\alpha$ in a higher precision would result in a better performance, as noted by R4.
>
> For more details about hyperparameters, please check Appendix C.
>
> (2) Related work
>
> Thanks for referring to [[YADIM](arxiv.org/abs/2009.00104)]. However, we strongly believe that there is no similarity between the main idea of this concurrent work (uploaded to arXiv in September) and ours. Their goals are completely different and orthogonal. While YADIM proposes a conceptual framework to categorize recent contrastive self-supervised learning works, AMDIM, CPC, and SimCLR, our work proposes a regularization method to improve contrastive self-supervised learning in a variety of settings. In other words, YADIM does not propose a regularization method for contrastive self-supervised learning like ours. Also, YADIM uses the encoder of AMDIM to extract multi-scale features, such that it might not be applicable to non-image domains; we confirmed that YADIM is only evaluated in the image domain. However, we may miss something, so it would be greatly appreciated if you could let us know which ideas in YADIM are similar to ours.
>
> (3) Code release
>
> Following R1's suggestion, we indeed provide the code and commands to replicate experimental results in another comment. As we mentioned in the abstract, the code will be publicly released to ensure reproducibility.

---

### Author Response · Authors · 2020-11-19
**Common Response to Reviewers**

Dear reviewers,

First of all, we appreciate your efforts in providing valuable comments on our paper. We've updated the manuscript following your suggestions. Here we summarize the changes:

1. We added the standard deviation of performances in tables.
2. We added ImageNet-100 results in Table 2, and replaced Figure 3 with linear classification results on the full ImageNet (previously, Figure 3 was evaluated on 10% of ImageNet for quick evaluation).
3. We added more description about InputMix in Section 3.3 and 4.4.
4. As suggested by R4, we replaced MoCo with MoCo v2 in Section 4 to avoid confusion.
5. We improved texts over the manuscript, following reviewers' suggestions.

We also shared the code in another comment, which is visible to PC, AC, and reviewers.

Please take a look and don't hesitate to remind us if we have missed something, or there are more suggestions.

Lastly we note that, as self-supervised learning is a timely topic, we are planning to discuss concurrent works in the final version. We are collecting them on our end, but it would be helpful if you share them as well.

Thank you very much!

Authors

---

### Decision · Program_Chairs · 2021-01-07
**Final Decision**

**Decision:**

Accept (Poster)

**Comment:**

Three reviewers recommended an acceptance (rating 7) while R1 deviated much from them (rating 3). After reading R1's concerns carefully and the authors' rebuttal, I found some of the criticisms to be invalid. The authors provided a satisfactory response, addressing concerns and clarifying potential misunderstandings. Because R1 did not update the review after the rebuttal period, I am assuming the concerns have been adequately addressed. The three other reviewers all unanimously agreed that this paper tackles a timely topic, proposes a simple and effective approach, and shows convincing empirical results. I concur with the reviewers' recommendations.